# TED-S: Twitter Event Data in Sports and Politics with Aggregated Sentiments

**Hansi Hettiarachchi** [1,*], **Doaa Al-Turkey** [1], **Mariam Adedoyin-Olowe** [1], **Jagdev Bhogal** [1] **and Mohamed Medhat Gaber** [1,2]

1   School of Computing and Digital Technology, Birmingham City University, Birmingham B4 7XG, UK;
    doaa.al-turkey@bcu.ac.uk (D.A.-T.); mariam.adedoyin-olowe@bcu.ac.uk (M.A.-O.);
    jagdev.bhogal@bcu.ac.uk (J.B.); mohamed.gaber@bcu.ac.uk (M.M.G.)
2   Faculty of Computer Science and Engineering, Galala University, Suez 435611, Egypt
*   Correspondence: hansi.hettiarachchi@mail.bcu.ac.uk

**Abstract:** Even though social media contain rich information on events and public opinions, it is impractical to manually filter this information due to data's vast generation and dynamicity. Thus, automated extraction mechanisms are invaluable to the community. We need real data with ground truth labels to build/evaluate such systems. Still, to the best of our knowledge, no available social media dataset covers continuous periods with event and sentiment labels together except for events or sentiments. Datasets without time gaps are huge due to high data generation and require extensive effort for manual labelling. Different approaches, ranging from unsupervised to supervised, have been proposed by previous research targeting such datasets. However, their generic nature mainly fails to capture event-specific sentiment expressions, making them inappropriate for labelling event sentiments. Filling this gap, we propose a novel data annotation approach in this paper involving several neural networks. Our approach outperforms the commonly used sentiment annotation models such as VADER and TextBlob. Also, it generates probability values for all sentiment categories besides providing a single category per tweet, supporting aggregated sentiment analyses. Using this approach, we annotate and release a dataset named *TED-S*, covering two diverse domains, sports and politics. TED-S has complete subsets of Twitter data streams with both sub-event and sentiment labels, providing the ability to support event sentiment-based research.

**Dataset:** https://github.com/HHansi/TED-S

**Dataset License:** Apache 2.0

**Keywords:** event detection; sentiment analysis; aggregated sentiments; Twitter; ensembled data annotation

## 1. Summary

Social media platforms generate increasing volumes of data with their growing popularity and widespread user bases [1]. These data contain a wide range of information together with public opinions important to diverse groups such as the scientific community and business world to capture exciting open challenges and get marketing decisions [2]. However, due to the vast volume and high dynamicity of data, it is impractical to analyse them manually to extract important events: incidents or activities which happened at a certain time and were discussed or reported significantly in social media [3] or their sentiments. Thus, many researchers focus on automated mechanisms to extract events and sentiments from social media data streams. To support them, different datasets with ground truth of events [4,5] and sentiments [6–9] have been published by previous work. However, to the best of our knowledge, no dataset contains both event and sentiment labels

together. Furthermore, a clear majority of the sentiment datasets consist of random sets of social media posts without the postings during a continuous period.

Considering these limitations and the importance of social media event sentiment analysis, in this paper, we release TED-S, a Twitter dataset with both sub-event and sentiment labels. We specifically targeted Twitter, considering its ideality for social network analysis based on popularity, simple data model and limited restrictions on data access [10]. Since events have different characteristics depending on the domain, we focus on data from two diverse domains, sports and politics, which have different sub-events, evolution rates, and audiences for this dataset. Deviating from traditional sentiment labelling, per tweet, we provide the probabilities/confidences for each sentiment category to further support aggregated sentiment analyses, such as analysing different sentiments expressed in a single tweet and the overall sentiment during a period or sub-event.

As our initial data source, we use *Twitter Event Data 2019 (TED)* [3] considering its recency and event coverage. TED consists of complete subsets of the Twitter data stream collected using Twitter Developer Standard API during the considered main events and their sub-event details extracted from published media reports. The considered main events are (1) *MUNLIV*—English Premier League 19/20 match between Manchester United Football Club (FC) and Liverpool FC on 20 October 2019, and (2) *BrexitVote*—Brexit Super Saturday 2019/ UK parliament session on Saturday, 19 October 2019. We assign sentiment labels for each tweet in these datasets (MUNLIV and BrexitVote) in our work to support future event sentiment research.

We target annotating the sentiment expressed by each tweet based on three categories as follows:

- **positive:** Hopeful, confident or expressing the good/positive aspect of a situation
- **negative:** Discouraging, refusing or expressing the bad/negative aspect of a situation
- **neutral:** No positive or negative expression

These categories are commonly used by previous research for Twitter sentiment analysis, considering their simplicity and coverage [8,9,11]. In addtion, since we target event data in our approach, using quantitative or more qualitative categories could introduce unnecessary complexities because the class boundaries could be thin, and definitions may need adjustments depending on the main event.

For sentiment labelling, different approaches have been used in previous research. Manual labelling is the most commonly used approach among them. In this approach, a group of annotators with sufficient background knowledge required for the targeted data do the labelling manually, and then a curator finalises the labels based on the majority opinion [6,7]. However, due to this process's time consumption and cost, its usage is mostly limited to the small dataset annotations. Overcoming these limitations, previous research has proposed different approaches, which can be mainly categorised as unsupervised lexicon-, distant supervision- and supervised machine learning-based approaches to generate large sentiment datasets.

- **Unsupervised Lexicon-based Approaches:** Considering the unavailability of pre-labelled data by the time of data labelling, there was a tendency to use unsupervised lexicon-based approaches for sentiment labelling. VADER [12–15] and TextBlob [8,14,16] were found to be the popularly used such unsupervised tools. VADER (Valence Aware Dictionary for sEntiment Reasoning) is a simple lexicon- and rule-based model designed for general sentiment analysis [17]. TextBlob is also a lexicon-based Python library designed for textual data processing covering a wide range of Natural Language Processing (NLP) tasks, including sentiment analysis [18]. However, due to the generic design of these tools, they fail to capture event-specific sentiment expressions accurately, as indicated by the results in Section 3.3.2.
- **Distant Supervision-based Approaches:** Distant supervision uses an existing knowledge base as a source to generate data labels, combining the benefits of semi-supervised and unsupervised approaches [19]. Go et al. [20] proposed using emoticons to derive sentiment labels of tweets following this technique. They recognised

emoticons which express positive and negative sentiments and labelled tweets which contain those accordingly. The same idea was also implemented using hashtags [21]. However, these approaches highly depend on the initial categorisation of emoticons/hashtags and are only capable of labelling tweets which have at least one of those emoticons/hashtags. In addition, they may require event-specific customisations to the original knowledge base depending on the targeted events.

- **Supervised Machine Learning-based Approaches:** With supervised learning, already available sentiment data or trained models were used to predict labels for large datasets [9,22]. These predictions are highly biased to the used training data and learning algorithm. Thus, similar to the scenario with unsupervised lexicon-based approaches, supervised approaches also will be less suitable for event-specific sentiment labelling. The comparatively low F1 scores we received during our experiments from the models only trained on available data further confirm this fact (Section 3.3.2).

Considering the limitations associated with the above approaches used by previous research, we propose a novel data annotation approach for sentiment labelling, following the idea of supervised machine learning-based approaches. Rather than using a single model, we propose using an ensemble of multiple neural network models to mitigate individual model biases on final predictions. The models are specifically picked based on the recent trends in NLP to obtain more accurate predictions. In addition, we manually label small subsets from each large dataset and allow the models to learn the specifics of targeted events using that data while learning from a large labelled dataset available. The implementation of our approach is publicly available with the labelled data to support similar data annotation tasks. A comprehensive data description is available in Section 2, and the data labelling approach is further described along with the data statistics in Section 3. In summary, the contributions of this paper are as follows.

1. To the best of our knowledge, *TED-S* is the first dataset that contains Twitter data corresponding to particular events (two diverse events from the sports and political domain) throughout a continuous period with both sub-event and sentiment labels.
2. Along with data, an ensembled data annotation approach appropriate for large datasets is proposed involving multiple state-of-the-art neural network models, and its implementation is released to support similar data annotation tasks.
3. As sentiments, non-manual annotations made for complete datasets, which hold a combination of confidence values for all targeted sentiment categories, are released, providing the ability to customise the data for either direct or aggregated sentiment analysis. In addition, manual annotations made for small fractions of data, which hold the mainly expressed sentiment of a tweet, are released to support future research avenues such as analysing manual and non-manual annotations and designing semi-supervised learning approaches, which strengthen the models by iteratively learning manual and predicted labels.
4. Availability of both event and sentiment labels unlike the existing datasets makes *TED-S* beneficial for a wide range of research in social media data, including event detection, sentiment classification, sentiment evolution, event sentiment extraction and event sentiment forecasting.

## 2. Data Description

In this section, we provide descriptions for all folders and files in our repository.

### 2.1. Folder: Data

This folder consists of all labelled data generated using the manual and non-manual approaches. All files in the folder are summarised in Table 1 and their attributes (column names) are described in Table 2.

**Table 1.** Descriptions for files in *data* folder.

| Folder | File | Description |
|---|---|---|
| manual | munliv_subset.xlsx | A random subset of MUNLIV data with manual sentiment annotations |
| | brexitvote_subset.xlsx | A random subset of BrexitVote data with manual sentiment annotations |
| non-manual | munliv_15.28-17.23.xlsx | MUNLIV data from 15:28 to 17:23 on 20/10/2019 with ensembled sentiment annotations |
| | munliv_15.28-17.23_no-duplicates.xlsx | MUNLIV non-duplicate data from 15:28 to 17:23 on 20/10/2019 with ensembled sentiment annotations |
| | brexitvote_08.00-13.59.xlsx | BrexitVote data from 08:00 to 13:59 on 19/10/2019 with ensembled sentiment annotations |
| | brexitvote_08.00-13.59_no-duplicates.xlsx | BrexitVote non-duplicate data from 08:00 to 13:59 on 19/10/2019 with ensembled sentiment annotations |

**Table 2.** Descriptions for data attributes.

| Attribute | Description |
|---|---|
| id | Tweet ID |
| label | Sentiment label (positive, negative or neutral) |
| timestamp | Tweet posted timestamp (format: yyyy-mm-dd hh:mm:ss) |
| positive_mean | Mean of the positive confidence values predicted by each model |
| negative_mean | Mean of the negative confidence values predicted by each model |
| neutral_mean | Mean of the neutral confidence values predicted by each model |
| positive_std | Standard deviation of the positive confidence values predicted by each model |
| negative_std | Standard deviation of the negative confidence values predicted by each model |
| neutral_std | Standard deviation of the neutral confidence values predicted by each model |

*2.2. Folder: Algo*

This folder consists of all the implementations of algorithms (or machine learning models) we used for ensembled annotations. More details about the models we used (LSTM, CNN and Transformer) are described in Section 3. An overview of the files in this folder is provided in Table 3.

**Table 3.** Descriptions for files in *algo* folder.

| Folder | File | Description |
|---|---|---|
| models/nn | nn_architecture.py nn_args.py nn_model.py nn_util.py | Python scripts for the implementations of neural network architectures (LSTM and CNN), arguments, model building flow and utilities |
| models/transformer | transformer_args.py transformer_model.py transformer_util.py | Python scripts for the implementations of transformer model, arguments and utilities |
| util | data_processor.py evaluate.py file_util.py label_encoder.py | Python scripts for the general utilities required by models: data preprocessing, model evaluation, file handling and label processing |

*2.3. Folder: Experiments*

This folder consists of Python scripts we used to execute all the experiments associated with the proposed annotation process as summarised in Table 4.

**Table 4.** Descriptions for files in *experiments* folder.

| File | Description |
| --- | --- |
| lstm_config.py<br>lstm_experiment.py | Python scripts for LSTM model-based experiments |
| cnn_config.py<br>cnn_experiment.py | Python scripts for CNN model-based experiments |
| transformer_config.py<br>transformer_experiment.py | Python scripts for Transformer model-based experiments |

## 3. Methods

This section presents our data annotation approach. Section 3.1 details the Twitter data collection we used. Section 3.2 describes the manual data annotation approach we followed to label subsets from each large dataset to utilise during the model learning process to provide specifics of targeted events to the models. Finally, Section 3.3 describes the ensembled approach proposed for data annotation, involving several neural network architectures, conducted experiments and data statistics.

### 3.1. Twitter Data Collection

We used *Twitter Event Data 2019 (TED)* (Twitter Event Data 2019 is available on https://github.com/HHansi/Twitter-Event-Data-2019 (accessed on 25 November 2021)) [3] as our initial data source to acquire event-related tweets. To the best of our knowledge, this is the most recent social media dataset released with ground truth (GT) event details covering two diverse domains, sports and politics. In addition, this dataset has complete subsets of Twitter data stream collected using Twitter Developer Standard API during the considered event periods without any gaps. The sports dataset was generated focusing on the English Premier League 19/20 match between Manchester United FC and Liverpool FC on 20 October 2019, and we refer to this dataset as *'MUNLIV'* similar to the original study. The political dataset was generated focusing on the Brexit Super Saturday 2019, a UK parliament session that occurred on Saturday, 19 October 2019, and it is referred to as *'BrexitVote'*. The statistics of the collected data are summarised in Table 5.

**Table 5.** Statistics of the collected tweets corresponding to the events: MUNLIV and BrexitVote.

| Dataset | Period (UTC) | Total Tweets | Non-Duplicates |
| --- | --- | --- | --- |
| MUNLIV | 15:28–17:23 | 99,837 | 41,721 |
| BrexitVote | 08:00–13:59 | 174,078 | 35,541 |

### 3.2. Manual Annotation

We targeted annotating random subsets from MUNLIV and BrexitVote datasets using the manual annotation process. We considered three sentiment categories: *positive*, *negative* and *neutral* for our annotation process following the definitions stated in Section 1. We targeted assigning the most appropriate category for each tweet based on its textual content. In cases when the main text has a conflicting sentiment with the hashtags, we gave priority to the main text because such hashtags are mostly used to connect the tweet to a particular topic.

We involved two annotators for this task who at least have a master's level qualification in computer science or linguistics. The annotators familiarised themselves with the targeted events before starting the annotation process by reading available resources. We also provided them with sample annotations per category to be familiar with the task. All annotators worked on the same 150 samples during the first annotation round to measure their inter-annotator agreement. The outputs of this round indicated 0.7393 and 0.6367 Cohen's kappa [23] between the annotations for MUNLIV and BrexitVote samples, respectively. Considering the high agreement achieved, we then annotated the remaining samples from both selected subsets using one annotator per instance. Completing the

manual annotation process, we obtained 8344 labelled tweets from the MUNLIV dataset and 2016 labelled tweets from the BrexitVote dataset. The distribution of the labelled tweets among the three sentiment categories is summarised in Table 6.

**Table 6.** Statistics of the manually annotated tweets corresponding to the events: MUNLIV and BrexitVote.

| Dataset | Negative | Neutral | Positive | Total |
|---|---|---|---|---|
| MUNLIV | 3488 | 1545 | 3311 | 8344 |
| BrexitVote | 840 | 289 | 887 | 2016 |

*3.3. Ensembled Annotation*

Considering the cost associated with a manual process in annotating large datasets and limitations in non-manual approaches on event-specific sentiment annotation, we propose an ensembled approach to annotate complete MUNLIV and BrexitVote datasets. For this approach, we utilise the data annotation strategy proposed with democratic co-learning [24] considering its successful applications in different areas such as time-series prediction [25] and offensive language identification [26]. In this approach, a set of classifiers are trained on available labelled data using different learning algorithms. When different algorithms with different inductive biases are involved, it helps resolve individual model biases and produces predictions with lower noise. Then, the trained models are used to make predictions on unlabeled data and aggregated the outputs to generate final labels.

Since our annotation task focuses on three sentiment categories: *positive, negative* and *neutral*, we build multi-class classification models with the ability to predict the confidence of each category, given an instance. Then, we aggregate the confidence values predicted by each model to generate the labels for unlabelled data. As the final label, we provide the mean and standard deviation of the confidence values predicted by each model per category rather than providing an exact category. Given a more detailed label with these values, the users have the potential to adjust data based on targeted applications. The standard deviation values will be specifically useful for filtering out instances with high model disagreement to reduce the noise in the dataset depending on the user requirements. In addition, providing confidence values for each sentiment category is helpful in scenarios when a single instance contains a mix of sentiments. A summary of our approach is as follows. It is also illustrated in Algorithm 1 in a more detailed manner.

1.  Train N diverse supervised models $M_j | j \in [1, N]$ using available labeled data to predict the sentiment categories $S = \{positive, negative, neutral\}$.
2.  For each instance $x$ in the unlabelled data, predict the confidence for each category $\{cf_i | i \in S\}$ using each built model $M_j$.
3.  Aggregate the predicted confidences per category of each instance $x$ to generate final label $l = (\{mean\text{-}cf_i | i \in S\}, \{std\text{-}cf_i | i \in S\})$ where $mean\text{-}cf_i = mean(\{cf_{i(M_j)} | j \in [1, N]\})$ and $std\text{-}cf_i = standard\text{-}deviation(\{cf_{i(M_j)} | j \in [1, N]\})$.

---

**Algorithm 1** Ensembled Annotation

---

1　$D \leftarrow$ unlabelled data;
2　$M \leftarrow []$;
　　// Train N diverse models
3　**for** *j=1 to N* **do**
4　　$M_j \leftarrow$ trained model using $algorithm_j$;
5　　$M.add(M_j)$;
6　**end**
　　// Generate labels per instance
7　**for** *x in D* **do**
8　　$cfs_{pos} \leftarrow []$; // Initialise a list to keep positive confidence values
9　　$cfs_{neu} \leftarrow []$; // Initialise a list to keep neutral confidence values
10　　$cfs_{neg} \leftarrow []$; // Initialise a list to keep negative confidence values
　　　// Predict confidence for each category using each built model
11　　**for** *j=1 to N* **do**
12　　　$\{cf_{pos}, cf_{neu}, cf_{neg}\} \leftarrow M_j.predict(x)$;
13　　　$cfs_{pos}.add(cf_{pos})$;
14　　　$cfs_{neu}.add(cf_{neu})$;
15　　　$cfs_{neg}.add(cf_{neg})$;
16　　**end**
　　　// Calculate mean of predicted confidences per category
17　　$mean\text{-}cf_{pos} \leftarrow mean(cfs_{pos})$;
18　　$mean\text{-}cf_{neu} \leftarrow mean(cfs_{neu})$;
19　　$mean\text{-}cf_{neg} \leftarrow mean(cfs_{neg})$;
　　　// Calculate standard deviation of predicted confidences per category
20　　$std\text{-}cf_{pos} \leftarrow standard\text{-}deviation(cfs_{pos})$;
21　　$std\text{-}cf_{neu} \leftarrow standard\text{-}deviation(cfs_{neu})$;
22　　$std\text{-}cf_{neg} \leftarrow standard\text{-}deviation(cfs_{neg})$;
　　　// Format final label
23　　$l \leftarrow (\{mean\text{-}cf_{pos}, mean\text{-}cf_{neu}, mean\text{-}cf_{neg}\}, \{std\text{-}cf_{pos}, std\text{-}cf_{neu}, std\text{-}cf_{neg}\})$;
24　**end**

---

The rest of this section explains the used learning algorithms/models, model evaluations and a summary of final labels. Under model evaluation, we summarise the details of training and testing data we used, obtained results and criteria we used to select the best models to label unlabelled data.

### 3.3.1. Models

The supervised sentiment analysis approaches developed by previous research range from traditional machine learning (ML) [27,28] to deep learning (DL) [29–31]. However, more focus has been given to DL-based methods in recent research, considering their improved performance over traditional ML-based approaches [10,32]. Among different DL methods, Long Short-term Memory (LSTM) [33] and Convolutional Neural Network (CNN) [34] were found to be the commonly used algorithms for sentiment analysis [35]. Also, transformer models [36] have been recently involved in sentiment analysis along with their success in several NLP applications [37–40]. Following these trends, we constructed three classification models based on LSTM, CNN and Transformer architectures, which have diverse inductive biases to use with the ensembled annotation approach.

LSTM:

LSTM model consists of five layers. First is an embedding layer initialised with concatenated GloVe and fastText embeddings. We used GloVe's Common Crawl (840B tokens) 300-dimensional model (GloVe pre-trained models are available on https://nlp.

stanford.edu/projects/glove/ (accessed on 28 December 2021)) and fastText's Common Crawl (with subword information) 300-dimensional model (fastText pre-trained models are available on https://fasttext.cc/docs/en/english-vectors.html (accessed on 28 December 2021)) to generate the embeddings. We experimented with separate embeddings and their concatenation in initial experiments, and concatenation performed best. Following the embedding layer, this architecture has two bi-directional LSTM layers with a dense layer on top. Finally, a dense layer with softmax activation is used to generate the predictions. We adapted this architecture from the *Toxic Comment Classification Challenge* in Kaggle (Toxic Comment Classification Challenge is available on https://www.kaggle.com/c/jigsaw-toxic-comment-classification-challenge (accessed on 28 December 2021)).

CNN:

CNN model consists of four 2D convolutional layers. Like LSTM, first is an embedding layer initialised with concatenated GloVe and fastText embeddings. Then, following a spatial dropout layer, this architecture has the convolutional layers, each with max-pooling layers. The Final is a dense layer with softmax activation to make predictions. We adapted this architecture from the *Quora Insincere Questions Classification* Kaggle competition (Quora Insincere Questions Classification is available on https://www.kaggle.com/c/quora-insincere-questions-classification (accessed on 28 December 2021)).

Transformer:

As the transformer model, we fine-tuned a pre-trained transformer for the sequence classification task [36]. Transformer models take a sequence as the input and return its representations as the output. The input sequence could contain one or two segments separated by a special token [SEP]. We used the one-segment scenario with no [SEP] tokens. As the first token of the sequence, another special token [CLS] should include, and it returns a special embedding representing the whole sequence which is used for text classification tasks. We pass this embedding through a softmax layer to generate the predictions. We adapted this architecture from the best system submitted for the *Multilingual Protest News Detection* shared task 1-English track in the workshop on Challenges and Applications of Automated Extraction of Socio-political Events from Text (CASE) 2021 [40]. For the transformer model, we used a variant of BERT, BERTweet [41] which is pre-trained on English tweets, considering its state-of-the-art results compared to other Twitter-specific models [11]. Among the available BERTweet models, we chose bertweet-base considering the targeted data volumes and used HuggingFace's Transformers library [42] to obtain the model.

3.3.2. Model Evaluation

This section describes the labelled data we used to train the models, model hyper-parameters, evaluation results and the best performing model selection for label generation. We used an Nvidia Tesla K80 GPU to conduct all our experiments.

Training Data:

In addition to the manually annotated data from MUNLIV and BrexitVote datasets, we used two available datasets: SemEval and FIFA, to train the models. SemEval dataset is obtained by merging all the datasets developed for SemEval sentiment analysis tasks from 2013 to 2016 [6]. This dataset consists of tweets corresponding to trending topics on Twitter covering different domains, including sports and politics. FIFA dataset consists of tweets corresponding to FIFA World Cup 2014 representing the sports domain [7]. The sentiment distribution of each dataset is illustrated in Figure 1. For model evaluation purposes, we split each of these datasets into two splits (train and test), and their details are summarised in Table 7. Additionally, the distribution of tweet sequence length in each split is shown in Figure 2. As can be seen in these graphs, recent tweets (MUNLIV and BrexitVote) tend to

have high sequence lengths than previous tweets (SemEval and FIFA) following the Twitter character limit increment.

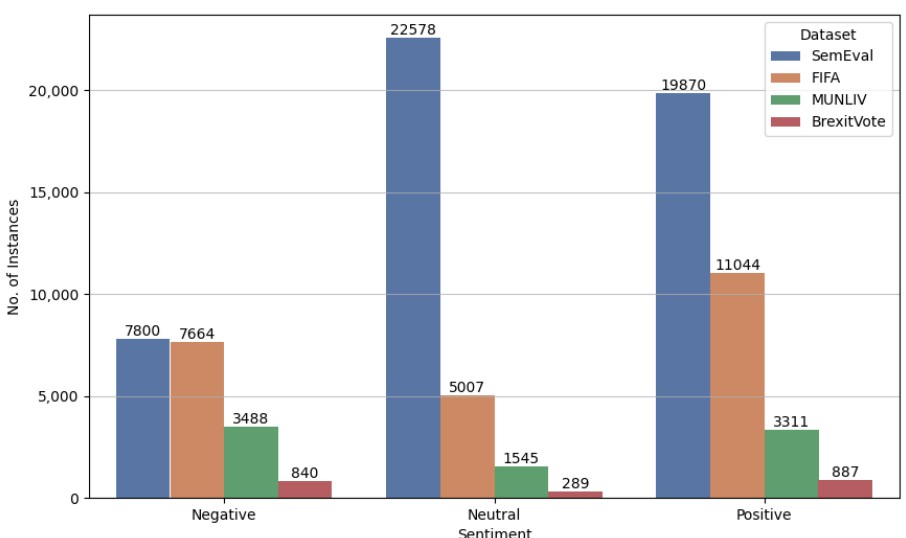

**Figure 1.** Sentiment distribution of labelled datasets selected to build models.

**Table 7.** Class distribution and data size of train and test splits.

| Domain | Dataset | Negative | Neutral | Positive | Total |
|---|---|---|---|---|---|
| General | SemEval-train | 6160 | 18,632 | 15,520 | 40,312 |
| | SemEval-test | 1640 | 3946 | 4350 | 9936 |
| Sports | FIFA-train | 6898 | 4506 | 9939 | 21,343 |
| | FIFA-test | 766 | 501 | 1105 | 2372 |
| | MUNLIV-train | 3070 | 1360 | 2914 | 7344 |
| | MUNLIV-test | 418 | 185 | 397 | 1000 |
| Politics | BrexitVote-train | 590 | 203 | 623 | 1416 |
| | BrexitVote-test | 250 | 86 | 264 | 600 |

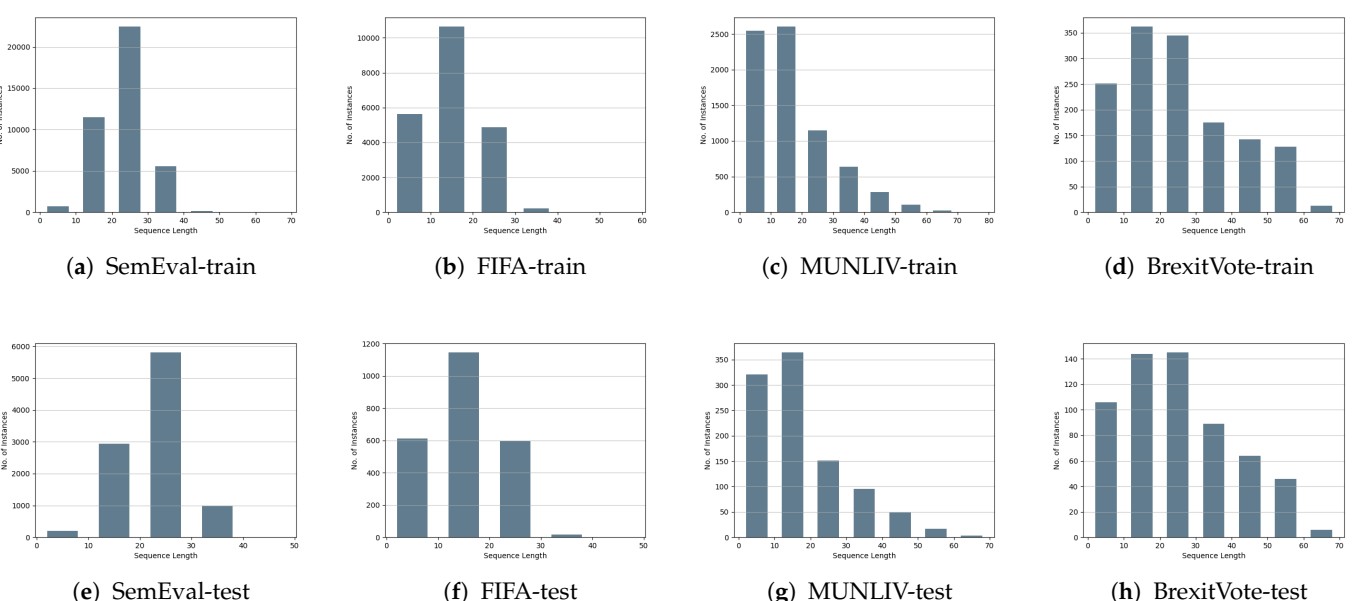

**Figure 2.** Sequence length histograms of train and test splits.

Data Preprocessing:

Under data preprocessing, we mainly considered cleaning uninformative tokens and formatting data according to model requirements. As uninformative data, we removed links and retweet notations in tweets. We tokenised data using Natural Language Toolkit (NLTK)'s (NLTK documentation is available at https://www.nltk.org/ (accessed on 28 December 2021)) TweetTokenizer with the 'reduce length' option, which generalises word forms by removing highly repeated characters. We converted text to lowercase for LSTM and CNN models because the used embedding models were trained on lowercased text.

Hyper-Parameters:

For all models, we fixed the max sequence length to 96, considering the sequence length distribution in datasets (Figure 2) and used 10% of the training data to validate the model during the training process. For LSTM and CNN models, we used the batch size of 64, a learning rate of $1e^{-3}$ with Adam optimizer and epochs of 20 with early stopping patience of 5. For the Transformer model, considering its high memory requirement, we used the batch size of 16, a learning rate of $1e^{-5}$ with Adam optimizer and 3 epochs with early stopping patience of 10. In addition, we set the evaluation steps allowing 6–11 evaluations per training epoch depending on the size of the dataset.

Results:

We evaluated the performance of each built model on all test datasets (Table 7) to select the best performing models. For training, we used separate as well as combined datasets. By combining datasets, we can increase data size and also analyse the inter-domain and inter-dataset capabilities in predicting the sentiment. When combining datasets, we ensure that each shares the same domain with at least one another set in combination to have sufficient instances from each domain in the final set. The Macro F1 score is used to measure model performance. The obtained results are summarised in Table 8.

Additionally, we evaluated the performance of previously proposed approaches for large dataset annotations to compare with the proposed approach. Among the three method categories we described in Section 1, we could only involve unsupervised lexicon- and supervised machine learning-based approaches for this comparison because distant supervision requires generating a knowledge base based on a specific data component (e.g., hashtag) and can only process the tweets which contain at least one of the defined components. As the unsupervised lexicon-based approaches, we used both VADER [17] and TextBlob [18] models. VADER returns a compound score within the range of $[-1, 1]$ which is commonly used for sentiment analysis. The extreme negative and positive scenarios are indicated by $-1$ and $1$. Following the common trend in previous research, we mapped compound scores $\geq 0.05$ to positive, $\leq -0.05$ to negative and the remaining to neutral sentiments during our evaluations [13,14]. Similarly, TextBlob also returns a polarity score within $[-1, 1]$, but negative, zero and positive values are recognised as negative, neutral and positive sentiments, commonly [14,16]. Their results are in the bottom section of Table 8. As the supervised machine learning-based models, we consider the models trained only using the other training data except for the data from the same dataset to which the test set belongs.

According to the results, in the majority of cases, the proposed neural network models outperformed the unsupervised lexicon-based approaches (VADER and TextBlob), indicating the effectiveness of supervision. Among these neural networks, mostly, the models trained only on other datasets (compared to the test data) returned comparatively low F1 measures compared to the models, which at least saw a fraction of the dataset to which the test set belongs, emphasising the importance of capturing event-specific sentiment expressions. For SemEval-test set, all the models trained using the combination of all training sets performed the best. For FIFA-test set, LSTM and BERTweet models trained on the combination of SemEval, FIFA and MUNLIV training sets and the CNN model trained on FIFA and MUNLIV training sets achieved the highest F1 values. The same LSTM

and BERTweet models performed best in MUNLIV-test set, but for CNN architecture, the model trained on the MUNLIV training set outperformed others. For BrexitVote-test set, different training combinations: LSTM trained on FIFA, CNN trained on SemEval, FIFA and MUNLIV, and BERTweet trained on FIFA and MUNLIV resulted in the best F1 values highlighting the inter-domain capabilities in predicting sentiments. Overall, LSTM and CNN models resulted in nearly similar F1 values, and the BERTweet model performed better than both of these models.

**Table 8.** Evaluation results of the models trained using different neural network architectures and unsupervised lexicon-based approaches (in italics). The best results are in bold.

| Model | Training Data | Macro F1 | | | |
|---|---|---|---|---|---|
| | | **FIFA-Test** | **SemEval-Test** | **MUNLIV-Test** | **BrexitVote-Test** |
| LSTM | FIFA | 0.6770 | 0.5399 | 0.6260 | **0.5667** |
| | MUNLIV | 0.5882 | 0.5302 | 0.6618 | 0.5014 |
| | SemEval | 0.6206 | 0.6185 | 0.5102 | 0.5156 |
| | BrexitVote | 0.3713 | 0.3338 | 0.3623 | 0.4034 |
| | FIFA+MUNLIV | 0.6583 | 0.5298 | 0.6739 | 0.5169 |
| | SemEval+FIFA+MUNLIV | **0.6781** | 0.6010 | **0.6807** | 0.5306 |
| | SemEval+BrexitVote | 0.6299 | 0.6093 | 0.5758 | 0.5592 |
| | SemEval+BrexitVote+FIFA+MUNLIV | 0.6685 | **0.6260** | 0.6786 | 0.5497 |
| CNN | FIFA | 0.6634 | 0.5513 | 0.6048 | 0.5686 |
| | MUNLIV | 0.5824 | 0.4887 | **0.6768** | 0.5090 |
| | SemEval | 0.6124 | 0.6034 | 0.5139 | 0.4998 |
| | BrexitVote | 0.4672 | 0.3779 | 0.4238 | 0.4734 |
| | FIFA+MUNLIV | **0.6724** | 0.5785 | 0.6657 | 0.5663 |
| | SemEval+FIFA+MUNLIV | 0.6551 | 0.6199 | 0.6622 | **0.5791** |
| | SemEval+BrexitVote | 0.6168 | 0.6136 | 0.5636 | 0.5660 |
| | SemEval+BrexitVote+FIFA+MUNLIV | 0.6697 | **0.6222** | 0.6720 | 0.5478 |
| BERTweet | FIFA | 0.7254 | 0.6339 | 0.7027 | 0.6034 |
| | MUNLIV | 0.6531 | 0.5380 | 0.7030 | 0.5327 |
| | SemEval | 0.6650 | 0.7021 | 0.5811 | 0.5522 |
| | BrexitVote | 0.4821 | 0.4053 | 0.4525 | 0.4671 |
| | FIFA+MUNLIV | 0.7283 | 0.6559 | 0.7208 | **0.6046** |
| | SemEval+FIFA+MUNLIV | **0.7289** | 0.6974 | **0.7261** | 0.5450 |
| | SemEval+BrexitVote | 0.6756 | 0.7035 | 0.6191 | 0.5691 |
| | SemEval+BrexitVote+FIFA+MUNLIV | 0.7237 | **0.7127** | 0.7221 | 0.5823 |
| *VADER* | - | *0.5735* | *0.5454* | *0.4031* | *0.4836* |
| *TextBlob* | - | *0.5043* | *0.4776* | *0.3802* | *0.4337* |

Model Selection:

Among the trained models, we selected the best-performing model of each architecture to automatically generate the sentiment labels of unlabelled tweets in MUNLIV and BrexitVote. For this selection, we did not directly rely on MUNLIV- or BrexitVote-test F1 values, considering the smaller size of these datasets. We used a *Weighted F1* measure calculated combining either MUNLIV or BrexitVote and one of the already available dataset's test data (FIFA or SemEval). With MUNLIV, we combined FIFA results because both represent football events. With BrexitVote, we combined SemEval results as this dataset cover general topics, including politics. While calculating the weighted F1, we provided a high weight to the MUNLIV and BrexitVote because we are going to use the selected models to predict sentiments in their complete datasets (Equations (1) and (2)).

$$Weighted\ F1_{MUNLIV} = 0.33F1_{FIFA} + 0.67F1_{MUNLIV} \tag{1}$$

$$Weighted\ F1_{BrexitVote} = 0.33F1_{SemEval} + 0.67F1_{BrexitVote} \tag{2}$$

Table 9 shows the weighted F1 values achieved by each model. According to them, for MUNLIV predictions, we selected LSTM and BERTweet models trained on SemEval, FIFA and MUNLIV training sets, and the CNN model trained on a combination of all training

sets. For BrexitVote predictions, we selected the LSTM model trained on SemEval and BrexitVote training sets, CNN trained on SemEval, FIFA and MUNLIV training sets and BERTweet trained on all training sets.

**Table 9.** Weighted F1 measures computed for each model. The best results are in bold.

| Training Data | Weighted $F1_{MUNLIV}$ | | | Weighted $F1_{BrexitVote}$ | | |
|---|---|---|---|---|---|---|
| | LSTM | CNN | BERTweet | LSTM | CNN | BERTweet |
| FIFA | 0.6428 | 0.6241 | 0.7102 | 0.5578 | 0.5629 | 0.6135 |
| MUNLIV | 0.6375 | 0.6457 | 0.6865 | 0.5109 | 0.5023 | 0.5344 |
| SemEval | 0.5466 | 0.5464 | 0.6088 | 0.5495 | 0.5340 | 0.6017 |
| BrexitVote | 0.3653 | 0.4381 | 0.4622 | 0.3805 | 0.4419 | 0.4467 |
| FIFA+MUNLIV | 0.6688 | 0.6679 | 0.7233 | 0.5212 | 0.5703 | 0.6215 |
| SemEval+FIFA+MUNLIV | **0.6799** | 0.6598 | **0.7270** | 0.5539 | **0.5925** | 0.5953 |
| SemEval+BrexitVote | 0.5936 | 0.5811 | 0.6378 | **0.5757** | 0.5817 | 0.6135 |
| SemEval+BrexitVote+FIFA+MUNLIV | 0.6753 | **0.6712** | 0.7226 | 0.5749 | 0.5723 | **0.6253** |

### 3.3.3. Final labels

Using the best models selected, we predicted the labels for complete MUNLIV and BrexitVote datasets (Table 5). As the final label per instance we provide three mean values: *mean-cf_positive*, *mean-cf_negative* and *mean-cf_neutral* and three standard deviation values: *std-cf_positive*, *std-cf_negative* and *std-cf_neutral* computed based on confidences predicted by models.

We release labels for non-duplicate and all tweets in the selected event streams, considering the different use cases of sentiment analysis associated with social media. Without duplicate data, we can efficiently analyse the sentiment of ideas expressed. In social networks, people share others' information (on Twitter, we refer to this as retweeting) mostly to indicate an agreement. If we remove duplicates from a social media data stream, we remove such social aspect-based details associated with it. Thus, with duplicates, we can analyse public opinions and their evolution. For example, let us assume that there exists a tweet *p* with a positive sentiment and *n* with a negative sentiment posted during time *t*, and *p* is retweeted ten times while *n* does not. If we ignore the duplicates (retweets), we recognise one positive and negative opinion from the data. If we consider the duplicates, it indicates that the majority have a positive opinion.

Sentiment Distribution:

Using the predicted labels, we illustrate the distribution of sentiments of each event during the targeted period in Figures 3–6. Since more than one tweet can be posted during a particular time, we aggregated the repeated values and showed the mean with a 95% confidence interval in these line graphs.

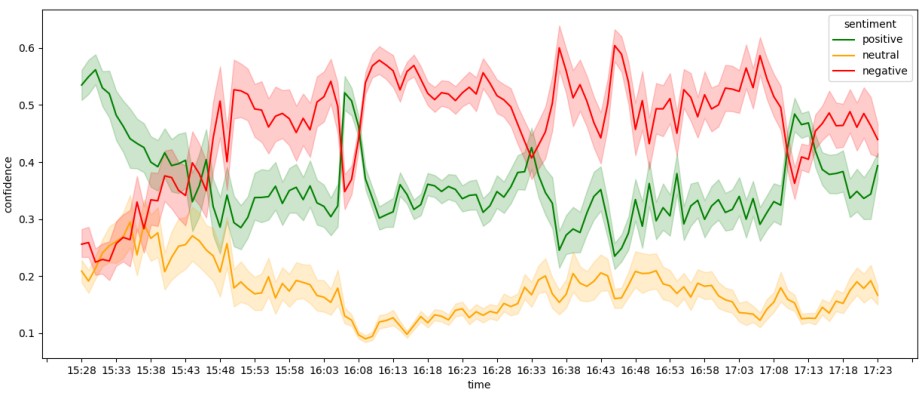

**Figure 3.** Sentiment distribution of MUNLIV tweets without duplicates on 20 October 2019 15:28–17:23.

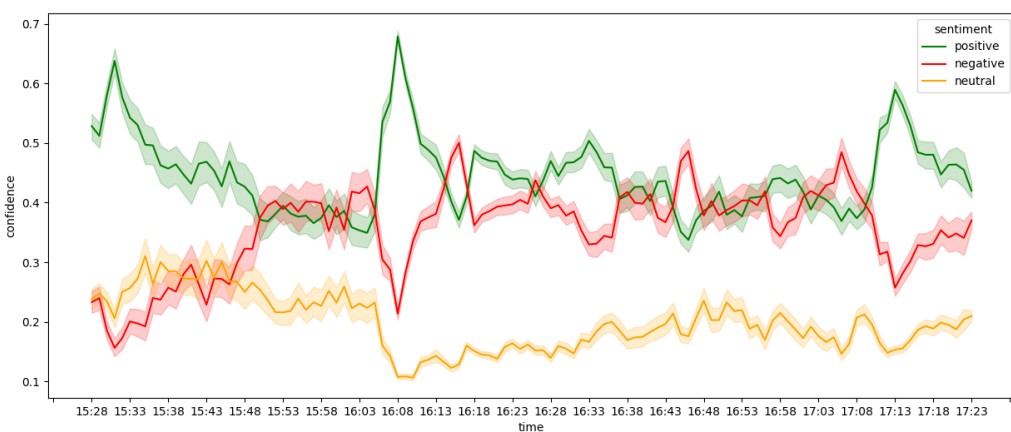

**Figure 4.** Sentiment distribution of MUNLIV tweets on 20 October 2019 15:28–17:23.

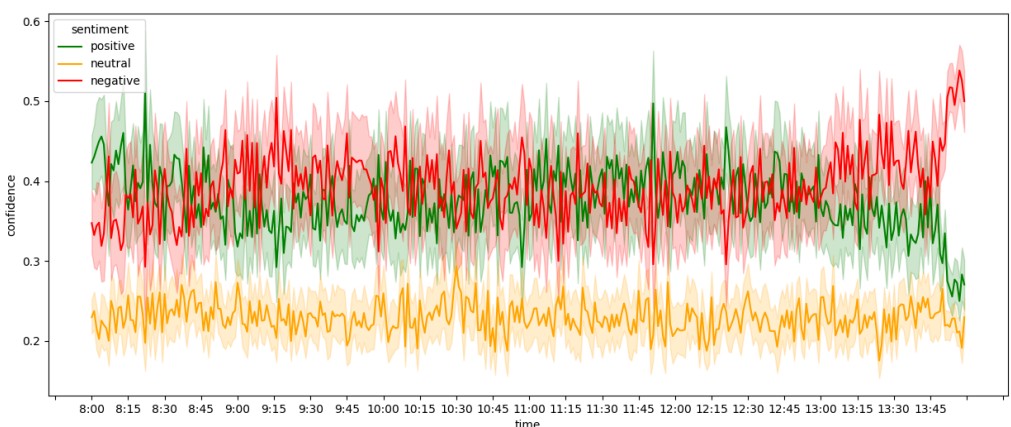

**Figure 5.** Sentiment distribution of BrexitVote tweets without duplicates on 19 October 2019 08:00–13:59.

For MUNLIV non-duplicate data, the majority expressed a negative sentiment, and the minority had a neutral sentiment (Figure 3). But, when we consider the sentiment distribution of all tweets, the majority is positive (Figure 4). This indicates that users retweeted most of the positive tweets during this event rather than writing their own. Additionally, the clear peaks in positive sentiment indicate the discussions that happened during or after the major sub-events of this football match (e.g., 16:06—first goal, 17:10—second goal). Similar to MUNLIV sentiment distribution, in BrexitVote also non-duplicate and all datasets show different positive and negative distributions. For non-duplicate data, there is a nearly similar positive and negative distribution throughout the considered time except at the end (Figure 5). In contrast to this, for all BrexitVote data, in the beginning, there was a combination of positive and negative opinions, but then the majority became negative, positive and negative again at the end (Figure 6).

Summary:

In this paper, we proposed a novel data annotation approach involving several neural network models, overcoming major limitations in available approaches for large dataset annotation, mainly the inability to capture event-specific sentiment expressions and the high impact of model biases. Using our approach, we assigned sentiment labels for all tweets (273,915 tweets) in TED [3], covering two main events: MUNLIV and BrexitVote in the domains of sports and politics. As the sentiment label per instance, we provide a composition of six values: means and standard deviations of confidences predicted per class (positive, negative and neutral) by built models. We release this new dataset under the name of TED-S, to be used with a wide range of research in social media, including event detection, sentiment classification and event sentiment extraction. In addition, we

release the implementation of our data annotation approach as an open-source project to support similar annotation tasks.

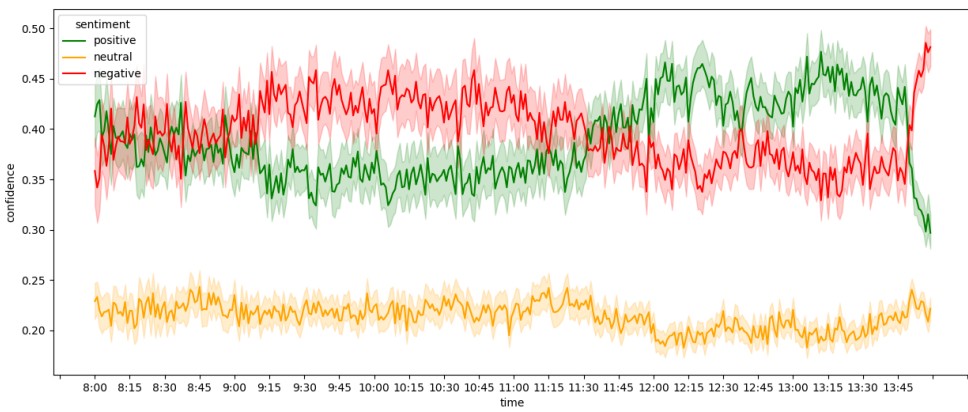

**Figure 6.** Sentiment distribution of BrexitVote tweets on 19 October 2019 08:00–13:59.

**Author Contributions:** Conceptualization, M.M.G. and H.H.; methodology, H.H.; software, H.H.; validation, H.H.; formal analysis, H.H. and D.A.-T.; investigation, H.H.; resources, H.H.; data curation, D.A.-T. and H.H.; writing—original draft preparation, H.H.; writing—review and editing, H.H., D.A.-T., M.A.-O., J.B. and M.M.G.; visualization, H.H.; supervision, M.M.G., M.A.-O. and J.B.; project administration, H.H. All authors have read and agreed to the published version of the manuscript.

**Funding:** This research received no external funding.

**Institutional Review Board Statement:** Not applicable.

**Informed Consent Statement:** Not applicable.

**Data Availability Statement:** The data presented in this study are openly available in https://github.com/HHansi/TED-S, accessed on 5 April 2022. Adhering to Twitter content redistribution policies, only the Twitter IDs are provided with the data, allowing to download of the full content from https://developer.twitter.com/en/docs/twitter-api/tweets/lookup/api-reference/get-tweets-id, accessed on 5 April 2022 (Twitter API).

**Conflicts of Interest:** The authors declare no conflict of interest.

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
