# Peer review of "TED-S: Twitter Event Data in Sports and Politics with Aggregated Sentiments"

_data_

Round 1
Reviewer 2 Report
It is an important manuscript and a valuable dataset for automated sentiment detection that other researchers could use.
My only concern is the performance of each model - even the best performing models did not go being ~ 0.7. Could the authors explain why it was not possible to achieve ~ 0.8-0.9 values?
Appreciate your comments.
Reviewer 3 Report
The work seems interesting despite being situated in a well-contested domain.
The focus of the two domains, sports and politics and thus the MUNLIV & BrexitVote sub-subsets, seems a bit restrictive although it's perfectly understandable that keeping the theme wide-open would probably produce fragmentation on each (naturally occurring) subset of domains.
Given the abundance of datasets based on tweets, contribution number 4 requires much more extensive support in order to stand. Also, given the (a) the importance of manual annotation for ground-truth estimation, and (b) abundance of non-manual annotation methods (and accordingly non-manually annotated datasets) the percentage of manually annotated tweets corresponding to the events: MUNLIV and BrexitVote seems low.
Please use a consistent tense in descriptions, e.g. "We involved two annotators for this task who at least have a master’s level qualification in computer science or linguistics. The annotators familiarise themselves with the targeted events before starting the annotation process by reading available resources."
The selection of the quantisation level (positive, negative and neutral) of the sentiment requires much more extensive discussion / support in order to stand - why not use a more detailed likert scale? is it enough for common applications? what do the rest of the dataset-oriented works choose?
The work does not present an equivalent to its size literature rereview / mention of related existing works, such mention would greatly support the work's motivation and contribution. Moreover, the work does indeed propose a semi-supervised data annotation approach appropriate for large datasets' sentiment annotation but does not truly compare results with state-of-the-art methods already published.
Round 2
Reviewer 1 Report
The authors have clarified the major concerns and the paper can be accepted at its current form.
Author Response
We appreciate your careful review and positive feedback. Your previous detailed comments have considerably helped to improve the manuscript to this state. We would like to thank you again for your feedback throughout this process.